# Opinions towards Medical Students’ Self-Care and Substance Use Dilemmas—A Future Concern despite a Positive Generational Effect?

**DOI:** 10.3390/ijerph192013289

**Published:** 2022-10-14

**Authors:** Paul McGurgan, Katrina Calvert, Elizabeth Nathan, Antonio Celenza, Christine Jorm

**Affiliations:** 1Division of Obstetrics and Gynaecology, University of Western Australia, Perth 6009, Australia; 2King Edward’s Memorial Hospital, Perth 6008, Australia; 3Division of Emergency Medicine, University of Western Australia, Perth 6009, Australia; 4Health and Medical Research Office, Australian Government Department of Health, Canberra 2601, Australia

**Keywords:** self-care, stress, medical students, doctors, public, drug/alcohol misuse

## Abstract

This study examines demographic factors which may influence opinions concerning medical students’ self-care and substance use behaviors as a means of providing insights into how future doctors view these issues compared to Australian doctors and members of the public. We conducted national, multicenter, prospective, on-line cross-sectional surveys using hypothetical scenarios to three cohorts- Australian medical students, medical doctors, and the public. Participants’ responses were compared for the different contextual variables within the scenarios and the participants’ demographic characteristics. In total 2602 medical students, 809 doctors and 503 members of the public participated. Compared with doctors and the public, medical students were least tolerant of alcohol intoxication, and most tolerant of using stimulants to assist with study, and cannabis for anxiety. Doctor respondents more often aligned with the public’s opinions on the acceptability of the medical students’ behaviors. Although opinions are not equivalent to behaviour, Australian students’ views on the acceptability for cannabis to help manage anxiety, and inappropriate use of prescription-only drugs are concerning; these future doctors will be responsible for prescribing drugs and managing patients with substance abuse problems. However, if current Australian medical student’s opinions on alcohol misuse persist, one of the commonest substance addictions amongst doctors may decrease in future.

## 1. Introduction

The 2017 World Medical Association’s decision to add ‘I will attend to my own health, well-being, and abilities in order to provide care of the highest standard’ to the Physicians’ Pledge [1] indicates the importance that doctors now place on this aspect of their contract with society [2]. Self-care is a broad concept; the World Health Organization defines it as ‘the ability of individuals, families and communities to promote health, prevent disease, maintain health, and to cope with illness and disability with or without the support of a healthcare provider’ [3].

Although occupational health and safety is a key component of medical education, data on medical students and qualified doctors paints a mixed picture regarding self-care and health [2,4,5,6,7,8,9,10,11]. Medical doctors appear to do well based on life expectancy metrics [5]. However, national and international studies reveal concerning levels of stress and burnout amongst medical students and doctors, associated with increased rates of mental illness, and maladaptive behaviours such as substance use and suicide [4,5,6,7,8,9,10,11,12,13,14].

The largest and most comprehensive study examining Australian doctors’ and medical students’ psychological wellbeing was undertaken by the mental health advocacy group, BeyondBlue [4]. This national survey used a variety of validated measures to detect psychological distress, alcohol/substance misuse, and burnout. Doctors and students reported higher rates of psychological distress, burnout and attempted suicide compared to both the Australian population, and other Australian professionals and students respectively. When doctors and medical students were asked about coping strategies if they felt anxious or depressed, more than 6% of doctors and 4% of students stated that they drank more alcohol, and approximately 1% and 0.6%, respectively, used non-prescribed medications in these circumstances. Medical student respondents disclosed that 10.2% used illicit drugs 2–3 times per month or less, with 0.5% stating at least weekly use [4].

The BeyondBlue results align with other national and international data on substance use. A large survey of US medical schools found that 91% of student participants reported using alcohol, 26% used cannabis, and 6% took amphetamines in a one-year period [6]. In 2018, a systematic review/meta-analysis on medical student cannabis use found that 8.8% of medical students were current users, with significant differences depending on the gender and population studied [10]. Recent surveys on the use of cognitive enhancing type drugs such as modafinil, found that 6% of Australian tertiary students (including medical students) reported using one or more prescription-restricted cognitive enhancing type drugs [15].

There is no consistent evidence demonstrating a direct relationship between substance use as a student and implications for future medical practice. However, Australian research found that more than 15% of doctors undergoing treatment for substance use problems had been abusing drugs since medical school [12]. Ayala et al.’s 2017 survey of US medical students showed that substance use by students had significant consequences, with more than 20% of student substance users reporting memory loss, and more than 10% admitting that they had driven while under the influence of alcohol or drugs [6].

Although there has been some research on the attitudes towards self-care training in medical schools [2], there is little published data comparing the public, medical students, and qualified doctors’ opinions on self-care/substance use dilemmas for medical students [16,17]. The Professionalism of Medical Students study (PoMS) is a body of research examining factors which influence opinions on medical students’ behaviors over a wide range of professionalism scenarios. Using validated surveys, the research sought to explore the effects of different contexts on professional decision making and assess whether demographic factors influenced opinions on the acceptability of behaviors [18]. Due to the known association between professionalism issues, burnout, and self-care [2,7,9], several PoMS scenarios specifically addressed self-care/substance use topics.

The aims of this study were to examine the nature and scale of factors which may influence opinions around medical student self-care/substance use related behaviors as a means of providing insights into how Australian doctors and medical students view these issues relative to members of the public, the implications of this for future health care professionals and those who may be involved in their care [19,20].

We sought to address the following questions:-do members of the public, qualified doctors and medical students have different opinions regarding medical student self-care/substance use dilemmas?-are respondents influenced by the seniority of the student protagonist (i.e., do respondents have different behavior expectations depending on whether a medical student is junior or senior in their course)?-does gender or age of the respondent influence their opinions on medical students’ behaviors?-are medical students’ opinions on self-care dilemmas influenced by what stage they are in their course?

## 2. Materials and Methods

### 2.1. Design, Ethics, Setting and Participants

This study was a national, multicenter, prospective, cross-sectional survey. Ethical approval was obtained from the University of Western Australia (UWA) for each of the cohorts: Australian enrolled medical students (HREC RA/4/1/8014), Australian medical doctors (HREC RA/4/1/9195), and Australian public (HREC RA/4/1/9278). We used a convenience sampling approach to survey medical students in Australia and New Zealand, members of the Australian public and Australian medical profession.

The methods and results of the medical student data, and validation of the survey instrument have been previously described in the Medical Students’ opinions on professional behaviors paper, referred to henceforth as the PoMS-I study [18]. As the student only data analysis indicated that national factors had an effect on opinions about students’ professional behaviors [18], for this paper we limited our study to Australian populations.

To optimize participation of a wide variety of Australian doctors and members of the public we utilized a range of approaches via conventional and social media. A social media webpage (Facebook Inc., Menlo Park, CA, USA) advertising the research project and link to the surveys was produced. To help notify medical doctors about the research project, the Australian Medical Association published information about the survey in their national news publication and distributed it through their General Practice News network. The Australian Confederation of Postgraduate Medical Education Councils sent the survey information to their listed directors of postgraduate medical education for dissemination through their networks.

For the public, in addition to the social media advertising, the Australian Consumers’ Health Forum placed information about the research project and survey on their website. The anonymous, online surveys were closed when recruitment had plateaued (less than 10 responses/month for 3 months) in April 2018.

### 2.2. Survey Instrument

The surveys used in the PoMS research covered a wide variety of professionally challenging situations pertinent to medical students [18]. Four of the scenarios related to a self-care issue. Most scenarios had two versions (vignettes) which contained a contextual variable identified a priori using the modified Delphi approach [18]. For example, the protagonist in the scenarios could be either a first year or final year medical student. The online survey (SurveyMonkey Inc., San Mateo, CA, USA) was designed such that each participant received a randomized version of each scenario to allow comparison of the contextual variables. This provided a means of analysing whether respondents were influenced by unconscious bias; that is, whether the seniority of the student protagonist influenced the respondents’ opinions on the acceptability of the behavior.

The scenarios were constructed to encourage participants to reflect on the acceptability of behaviors which ranged from serious professionalism breaches to positive examples of professional behavior. All scenarios ended with the sentence ‘This student’s behavior is…’. Respondents gave their opinions on ‘acceptability’ using a four-point Likert scale, with no option for equipoise.

Although the surveys were anonymous, participants were asked some questions about their background specifically gender, age (if a member of the public or a doctor), type of entry and year level in medical school (if a student). These questions (Appendix A) enabled the authors to check for participant contamination, for example, a member of the public completing the doctors’ on-line survey or vice versa. Post survey closure, the lead author checked for any participant allocation errors, and re-allocated these surveys (<10) to the appropriate survey cohort.

Medical students were classified as ‘junior’ if they were in the first two years of a post-graduate entry course or in years 1–3 for an undergraduate course; by default, senior students were in the latter years of these courses. The public and doctor surveys asked participants for their age; medical student surveys did not include this question, however the Medical Deans National Data Reports document that more than 95% of enrolled Australian medical students were less than 35 years old during the survey period [21].

The surveys are included in Appendix A survey scenarios relating to self-care themes along with their contextual variables are described in Table 1.

### 2.3. Data and Statistical Analysis

Likert scale responses were combined into binary categories of all ‘acceptable’ responses against all ‘unacceptable’ responses and summarized using frequency distributions. The survey design enabled the comparison of two types of variables- the first being the participants’ demographic group/sub-group background, for example, doctor, medical student, or member of the public/gender. The second type of variable was the randomized contextual variable within each version of the self-care scenarios, for example a junior or senior medical student protagonist using stimulant drugs to help them study.

The Chi-square or Fisher-Freeman-Halton test was used to compare responses for the different contextual variables within the scenarios and the participants’ demographic characteristics—student, doctor, member of the public, gender and age (if public or doctors)/seniority in the course (if a medical student). Logistic regression analysis was conducted separately for students, doctors and the public to assess possible differences in responses depending on gender, age (doctors and the public), and seniority for the student participants (junior or senior). Possible relationships (interactions) between the demographic characteristics were also assessed in the models for the relevant respondent groups with each scenario. Adjusted odds ratios (aOR) and 95% confidence intervals (CI) were presented. Null and blank responses were excluded for all questions, as were ‘prefer not to say’ responses for gender. SPSS Statistics 24 (SPSS Inc., IBM, Chicago, IL, USA) was used for data analysis and a *p*-value of <0.05 was considered statistically significant. Figures in tables provided may not add up to 100 per cent due to rounding (nearest decimal place) and totals may vary due to some participants not completing all of the questions, or for example, using option ‘prefer not to specify’ for gender question.

## 3. Results

The demographic details of the survey respondents are presented in Table 2. The participation of all Australian medical schools resulted in larger numbers of student respondents (*n* = 2602) compared to doctors (*n* = 809) or the public (*n* = 503). There was a significantly larger proportion of female participants in all three cohorts (*p* < 0.001). The three demographic groups’ surveys each contained specific questions regarding participant’s background to avoid participant contamination, for example, a member of the public inadvertently completing the doctors’ on-line survey or vice versa (Appendix A).

As data on medical student demographics recorded that >95% of medical students were under 35 years of age when the surveys were conducted [21], we stratified the public and qualified doctors into two age cohorts; ≤35 years of age, and >35 years of age, so that we could analyze the data for influence of age. Public respondents were significantly older than the qualified doctors who participated in the survey (*p* < 0.001).

Survey responses were analyzed to examine the nature and factors which may influence opinions on medical student self-care related behaviours and answer specific questions. As the scenarios on alcohol and drug misuse had contextual variables to assess if respondents were influenced by whether the protagonist was a junior or senior medical student, Table 3 illustrates the responses for each contextual variable, and the composite (combined total) response. Most respondent demographic groups (students, pubic or doctors) did not appear to be influenced by the seniority of the student when providing their opinions on the protagonist’s behavior. In view of this, the composite results were used in Table 4′s adjusted analysis for the influence of demographics on the acceptability of substance use (Table 4).

### 3.1. How Do Medical Students View Student Self-Care/Substance Use Dilemmas?

Overall, medical students had the lowest acceptability for alcohol intoxication (10.2%), and the highest acceptability for stimulant (12.2%) or cannabis use (19.7%) compared to public or doctor survey participants. Students were significantly more likely than doctors to consider it acceptable to misuse stimulant drugs, and significantly more likely than both doctors and the public, to consider it acceptable for a medical student to use cannabis to help manage anxiety (Table 3).

The only scenario that resulted in consistent opinions from all three cohorts surveyed involved the medical student protagonist seeking assistance; almost all respondents agreed on the acceptability of this behavior (Table 3). When comparing the acceptability of alcohol versus cannabis versus stimulants in the scenarios, it was notable that stimulant drug use had generally the least acceptability, whilst using cannabis to help with anxiety was regarded as being the most acceptable.

### 3.2. Do Demographic Factors Influence Opinions on the Acceptability of Self-Care/Substance Misuse Dilemmas?

Table 4 shows that respondents’ gender was the most pervasive factor influencing their opinions. This was most notable for the public; for each of the alcohol/drug use scenarios, male members of the public were significantly more likely to consider these as acceptable behaviors compared to female respondents. Conversely, the gender of the respondent appeared to have least effect on the opinions expressed by doctors.

The scenario which exhibited the most consistent gender effect was the use of stimulant drugs to assist with study. Male respondents, whether they were public, students or doctors, were typically at least twice as likely to consider stimulant drug use in this context as being acceptable.

When matched for respondents with an age under 36 years old, medical students had similar opinions to public and doctor participants for the stimulant and cannabis use scenarios. However, medical students were significantly less likely to consider alcohol intoxication as being acceptable. Older doctors were significantly less likely to consider it acceptable for the medical student protagonist to misuse alcohol or cannabis than younger doctors.

Medical students’ duration in the course (i.e., the influence of seniority) had notable effects (Table 4). Students in the senior years of medical study were significantly more likely to consider it acceptable to use stimulant drugs for study (aOR: 1.44, 95% CI 1.11–1.87, *p* = 0.007), cannabis for anxiety (aOR: 1.24, 95% CI 1.00–1.53, *p* = 0.05), and behave irresponsibly with alcohol (aOR: 1.68, 95% CI 1.27–2.23, *p* < 0.001).

## 4. Discussion

This is the first study to triangulate opinions on a national scale from the public, medical students, and qualified doctors on a range of self-care/substance use dilemmas involving medical students. It provides a unique insight into the factors which influence opinions on these issues, and the potential challenges in this area for future health professionals with known increased risk factors for burnout and psychological distress. The results demonstrate that whilst medical students have similar opinions on self-care dilemmas as members of the public and qualified doctors, they differ significantly in their opinions towards alcohol/substance use when compared to these other groups.

Regarding possible limitations in our study design, we used a convenience sampling survey methodology. Selection bias was minimized by using a wide range of methods to promote recruitment from the three cohorts. This included social media, news outlets and health consumer forums, and continuing the survey process until there were >500 participants in each demographic group. Although there is no denominator data for the doctor or public participants, PoMS-I calculated the medical student participation rate to be 15.2% of all Australian enrolled medical students, making it one of the largest medical student datasets in the literature [18]. As a result of confining the surveys to Australian populations, we cannot generalise that similar results would be obtained in other countries. The PoMS-I study which involved both Australian and New Zealand medical students, and other international studies [22] show that opinions on professional behaviours may be influenced by national cultural norms.

The surveys were designed to minimize respondents’ social desirability bias, the tendency for participants to understate their own negative behaviours. This was achieved by ensuring participants’ anonymity, and the scenarios being viewed through a 3rd party perspective. Although surveying participants’ opinions on the protagonist’s behaviors could be considered too simplistic an approach for the complex subject of self-care, many psychological studies of human behavior have used and validated similar approaches [17,23]. The caveat with this approach is that our results are limited to opinions on the acceptability of certain self-care behaviors, rather than indicating whether medical students or doctors would, or intend to engage in these behaviours.

As the WHO definition of self-care is wide-ranging, our surveys focused on specific self-care/substance use behaviors that are particularly relevant to the occupational health of medical students and future doctors, and had been examined in other studies [4,6,10,15,16,17]. It was reassuring to find extremely high levels of acceptability for a medical student seeking assistance after a traumatic event in all of the cohorts surveyed. However, some of the opinions expressed toward alcohol and substance use by various demographic groups, and the effect of participants’ gender and age influencing opinions merit further consideration. These are discussed under the following headings.

### 4.1. Demographic Groups’ Opinions Regarding the Medical Students’ Self-Care/Substance Use Dilemmas

Although longitudinal international studies have shown a consistent decline in alcohol consumption amongst younger people since 1995 [24,25] indicating a generational cohort effect [26]. Our finding that medical students’ have a significantly lower acceptability towards alcohol intoxication than their age matched public and doctor contemporaries (*p* < 0.001; Table 3), indicates that medical students view alcohol misuse differently to the other age-matched survey participants. Given that alcohol is currently the commonest substance abused by doctors [4,13,19], Australian medical students’ overwhelmingly negative opinions on intoxication may herald a future decline in the rate of alcohol abuse amongst doctors.

In contrast, when compared to the overall public and doctors’ opinions on cannabis and stimulant drug use, medical students were the cohort most likely to consider misuse of these substances as acceptable. Stimulant drug use is increasingly common in university students, including those studying health sciences [15,27]. Although concerns have been raised about the prevalence of stimulant drug use in medical students and doctors given their potential for abuse [15,27,28]. there have been some authors arguing the merits of these agents [29].

The differing opinions on stimulant drug use may reflect cultural norms. In 2015, the GMC surveyed UK medical students using a similarly worded scenario on stimulant drug use. They found that 8% of their students considered this acceptable behavior [17]. This is significantly different (*p* < 0.001) from the findings of our study, in which 12.2% of students considered it acceptable. Although the surveys do not explain why such differences in opinions may exist, they further support the literature that opinions on medical professional behaviors can be influenced by national cultural norms [18,22,30].

Cannabis is reported to be the most common illicit substance used by medical students, with an estimated 1 in 3 medical students reporting use [10]. Cannabis is increasingly promoted as an effective treatment for a wide range of conditions despite concerns about the limited evidence by which these claims are substantiated [31,32]. The promotion of the possible therapeutic benefits of cannabis may be one reason why doctors and medical students were the demographic groups with the highest acceptability for its use in managing anxiety.

The survey findings that doctor cohort respondents’ opinions more often aligned with those of the general public than those of medical students is notable as it is in contrast to the pattern of responses seen in other PoMS data examining opinions towards patient safety dilemmas [33]. It appears that opinions on self-care related to alcohol or drug use are not influenced by professional identity formation—the students adopting the norms of the behavior of the community they wish to identify with, but instead appear to be ‘generational’ in nature [26].

### 4.2. The Implications of Respondents’ Gender or Age Influencing Their Opinions on Medical Students’ Self-Care Behaviors

#### 4.2.1. Gender

The literature reveals that gender often has significant effects on both opinions towards, and actual self-care behaviors including substance use [4,6,10,13,19] and health professional suicide attempts [4,8]. The UK GMC study [17] found that in almost two thirds of their survey’s professionalism dilemmas, the respondents’ gender significantly influenced their opinion on the acceptability of the behavior. Likewise, the PoMS-I study [18] found that respondents’ gender was the most influential factor in determining whether a behavior would be described as acceptable (or not). Despite the association between gender and self-care behaviors, our results demonstrate that Australian doctors surveyed had no gender bias on their opinions towards cannabis use, and both Australian doctors’ and medical students had gender neutral norms for alcohol intoxication.

#### 4.2.2. Age

When medical student survey responses were compared with those of public and doctor participants under 36 years of age, medical students had similar opinions towards stimulant drug use and cannabis. This implies that opinions on the acceptability of misusing these substances are influenced by generational life cycle effects [26]. This was most notable with cannabis use; public and doctor participants were significantly more likely to express different opinions on the acceptability of cannabis use for anxiety if they were over 36 years old. This finding has been replicated in other national studies [34].

### 4.3. The Possible Consequences of Medical Students Viewing Self-Care Behaviors Differently as They Transition through Medical School

Although opinions are not equivalent to behaviors [35,36], the theory of planned behavior (TPB) has been used to understand and influence the nexus between students’ opinions and their behavior [18,30,36]. The theory states that an individual’s likelihood of engaging in a behavior is influenced by three factors—their attitudes towards the behavior, their perceptions of the social norms, and their perceived ability to perform the behavior [37].

The finding that medical students who were in the later stages of the course were more likely to consider cannabis/stimulant/alcohol misuse as being acceptable may be an age/life cycle effect due to the increased likelihood of encountering these situations as they progress through university. The PoMS-I paper found that more than 28% of medical students had encountered situations similar to the stimulant drug use scenario, 14.3% for alcohol intoxication, and 8.5% for cannabis [18]. Kulac and Franco both found a strong relationship between observation and participation in terms of unprofessional behaviour; medical students were more likely to perceive unprofessional behaviors as being ‘acceptable’ if the student had either observed or participated in the activity [30,36].

Medicine has a long history of doctors’ experimenting with substances, for example the use of ether. Medical students’ opinions about substance use behaviors and self-prescribing have been shown to influence their approach to patient counselling [38,39] and reporting impaired colleagues [40]. Hence, whilst it is reassuring to note the comparatively low acceptability of medical students towards alcohol intoxication, their tolerant views on medical student use of other substances may be problematic. This is particularly relevant given the concerns noted in the Introduction on the possible medium- and long-term implications for medical students using drugs/alcohol, and underscores the importance of further qualitative research in this area to gain a better understanding of the basis for medical students’ attitudes.

## 5. Conclusions

Despite self-care for health professionals and students in these disciplines receiving more attention [2,4], there is little published data on opinions regarding medical student self-care behaviors. This paper demonstrates that opinions related to alcohol and substance misuse by medical students are influenced by a variety of factors, some of which are inter-related:
Medical students have significantly different opinions towards alcohol/substance use than doctors or members of the public;Opinions on medical student substance use suggest that generational influences rather than health professional cultural norms determine the acceptability of behaviours;Student respondents in the latter stages of their medical course were significantly more likely to consider either stimulant drug use or taking cannabis as being acceptable;In comparison with other international studies, Australian medical students were more likely to consider it acceptable to use stimulant drugs to assist with study;Medical students and doctors appear to have gender neutral norms for alcohol intoxication.

Although opinions are not the same as actions, as future health care providers/medication prescribers with known risk factors for burnout and psychological distress, Australian medical students’ views on the acceptability for cannabis to help manage anxiety, and use of prescription-only drugs to help study are a concern. However, if Australian medical student’s opinions on alcohol persist, the most prevalent current substance addiction amongst doctors may decrease.

## Figures and Tables

**Table 1 ijerph-19-13289-t001:** PoMS Survey Scenarios Relating to a Self-care Theme with the Contextual Variables.

Theme	Contextual Variable	Scenario [Alternative Version with Contextual Variable]
Seeking assistance	N/A	A medical student rushes to an emergency bell on the ward. They are the first responder and commence effective CPR. Despite the arrival and assistance of the Medical Emergency Team the patient dies. The student is distressed by recurring thoughts of the event, which affect their sleep. When these symptoms continue, the student seeks assistance from Student Support Services. This student’s behavior is:
Alcohol misuse	Medical student seniority	The Medical School is informed that a first year [final year] medical student has been charged with drunk and disorderly conduct after an altercation at a night club. This student’s behavior is:
Using illicit drugs to assist with study	Medical student seniority	During exam time a first year [final year] medical student buys stimulant drugs online that are usually only available on prescription, and uses them in order to stay awake and study. This student’s behavior is:
Using cannabis to treat anxiety	Medical student seniority	A first year [final year] medical student in an undergraduate entry medical course fails all of their end of year examinations at the first attempt. The student had not applied for any special consideration, but subsequently admits that their ability to study had been affected by anxiety and they had used cannabis daily to provide relief over the past 2 months. This student’s behavior is:

**Table 2 ijerph-19-13289-t002:** Demographic Details of PoMS Survey Participants: Gender/Age/Medical Students’ Seniority ^a^.

Demographic Group (Total Number)	Gender	
	Male Participants (%)	Female Participants (%)
Public (503)	147 (29.2)	349 (69.4)
Medical doctors (809)	200 (24.7)	599 (74)
Medical students (2602)	1105 (42.8)	1439 (55.8)
**Demographic Group** (Total number)	**Age ^a^**	
	≤35 years of age (%)	>35 years of age (%)
Public (503)	164 (32.8)	336 (67.2)
Medical doctors (809)	390 (48.4)	416 (51.6)
**Demographic Group** (Total number)	**Students’ stage in the medical course ^a^**	
	‘Junior’	‘Senior’
Students’ stage in course ^a^ (2602)	1214 (50.6)	1184 (49.4)

Notes. Total numbers in tables may have discrepancies due to some participants not completing all demographic questions. ^a^ Public and Doctor surveys asked participants for their age. These data were not available for medical students. Medical students were classified as ‘junior’ if they were in the first two years of a post-graduate entry course or in years 1–3 for an undergraduate course; by default, senior students were in the latter years of these courses.

**Table 3 ijerph-19-13289-t003:** Effect of Participants’ Demographic Backgrounds on their Responses to the PoMS Self-care Scenarios with Contextual Variables (Seniority of the Student Protagonist) Included and Combined Total Responses.

Scenario	Contextual Variable	Public *n*/*N* (%) ^b^	Doctors *n*/*N* (%) ^b^	Students (*n*/*N*) % ^b^	Global Test (Students vs. Public vs. Doctors) *p*-Value ^c^	Pairwise Comparison between Demographic Groups *p*-Value ^d^
	Combined Total Response ^a^					
Seeking assistance	N/A	440/450 (97.8)	729/731 (99.7)	2243/2256 (99.4)	0.001	Public vs. Students < 0.001 Public vs. Doctors = 0.002
Alcohol misuse	Junior student	35/244 (14.3)	67/361 (18.6)	133/1130 (11.8)		
	Senior student	35/205 (17.1)	45/365 (12.3)	96/1126 (8.5)		
	Combined ^a^	70/449 (15.6)	112/726 (15.4)	229/2256 (10.2)	<0.001	Students vs. Public < 0.001 Students vs. Doctors < 0.001
Using illicit drugs to assist with study	Junior student	19/231 (8.2)	26/330 (7.9)	136/1131 (12.0)		
	Senior student	28/218 (12.8)	36/393 (9.2)	138/1114 (12.4)		
	Combined ^a^	47/449 (10.5)	62/723 (8.6)	276/2258 (12.2)	0.022	
Using cannabis to treat anxiety	Junior student	24/223 (10.8)	70/348 (20.1)	206/1099 (18.7)		
	Senior student	33/224 (14.7)	45/370 (12.2)	236/1143 (20.6)		
	Combined ^a^	57/447 (12.8)	115/718 (16.0)	442/2242 (19.7)	<0.001	Students vs. Public < 0.001 Students vs. Doctors = 0.027

Notes. Total numbers in tables may have discrepancies due to some participants not completing all demographic questions. ^a^ Combined = sum of both junior and senior student vignette responses. N/A if no contextual variable in the Scenario. ^b^ (*n*) number of respondents stating either ‘Acceptable/mostly acceptable’ opinions on protagonist’s behavior/ (*N*) Number of respondents and (%). ^c^ Chi-square tests used for binary variables (Combined Acceptable/Mostly Acceptable vs. Combined Unacceptable/Mostly Unacceptable responses) and the global p-value is reported in each scenario, unless expected count value <5 in which case the Fisher-Freeman-Halton Exact test p-value is reported. ^d^
*p*-values < 0.017 considered statistically significant after Bonferroni adjustment for 3 pairwise comparisons (alpha = 0.05/3).

**Table 4 ijerph-19-13289-t004:** Adjusted Analysis of Participants’ Demographics on their Opinions for the Acceptability for Alcohol/Substance Misuse by a Medical Student.

Substance	Demographic Group	Respondents’ Gender Male *n*/*N* (%) vs. Female *n*/*N* (%); aOR (95% CI); [*p*-Value] ^b^	Respondents’ Age ≤aOR (95% CI); [*p*-Value] ^b^	Medical Student Respondents’ Seniority ^a^ Junior *n*/*N* (%) vs. Senior *n*/*N* (%); aOR (95% CI); [*p*-Value] ^b^
Alcohol	Public	28/129 (21.7) vs. 40/316 (12.7); 1.88 (1.1–3.22); [0.021]	29/147 (19.7) vs. 41/302 (13.6); 1.52 (0.89–2.6); [0.123]	
	Doctors	26/177 (14.7) vs. 86/542 (15.9); 0.98 (0.61–1.59); [0.943]	65/349 (18.6) vs. 47/377 (12.5); 1.63 (1.08–2.46); [0.02]	
	Students	94/958 (9.8) vs. 133/1285 (10.4); 0.92 (0.7–1.22); [0.56]		84/1092 (7.7) vs. 145/1172 (12.4); 1.68 (1.27–2.23); [<0.001]
Stimulants	Public	26/128 (20.3) vs. 21/317 (6.6); 3.58 (1.91–6.69); [<0.001]	25/147 (17) vs. 22/302 (7.3); 2.62 (1.4–4.89); [0.003]	
	Doctors	22/177 (12.4) vs. 40/539 (7.4); 1.88 (1.08–3.29); [0.026]	34/347 (9.8) vs. 28/376 (7.4); 1.47 (0.86–2.49); [0.157]	
	Students	160/945 (16.9) vs. 110/1278 (8.6); 2.12 (1.63–2.75); [<0.001]		108/1078 (10) vs. 166/1167 (14.2); 1.44 (1.11–1.87); [0.007]
Cannabis	Public	31/127 (24.4) vs. 24/316 (7.6); 3.92 (2.18–7.05); [<0.001]	29/147 (19.7) vs. 28/300 (9.3); 2.39 (1.33–4.31); [0.004]	
	Doctors	33/177 (18.6) vs. 82/534 (15.4); 1.39 (0.89–2.19); [0.152]	69/346 (19.9) vs. 46/372 (12.4); 1.85 (1.23–2.79); [0.003]	
	Students	216/951 (22.7) vs. 224/1278 (17.5); 1.37 (1.11–1.69); [0.004]		192/1076 (17.8) vs. 250/1166 (21.4); 1.24 (1.00–1.53); [0.05]

Notes. Total numbers in tables may have discrepancies due to some participants not completing all demographic questions. Logistic regression analyses were conducted separately for each demographic group. Models for the public and doctors were adjusted for gender [male vs. female (referent)] and age [≤35y vs. >35y (referent)]; models for the medical students were adjusted for gender and seniority [senior vs. junior (referent)]. ^a^ Senior students defined as either undergraduate entry years 4–6 or post-graduate entry years 3–4. Junior students defined as either undergraduate entry years 1–3 or post-graduate entry years 1–2. ^b^ aOR = Adjusted Odds Ratio; CI = Confidence Interval.

## Data Availability

Not applicable.

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
