# Peer review of "Opinions towards Medical Students’ Self-Care and Substance Use Dilemmas—A Future Concern despite a Positive Generational Effect?"

_ijerph, 2022, doi:10.3390/ijerph192013289_

Round 1

Reviewer 1 Report

This manuscript entitled “Opinions Towards Medical Students’ Self-care and Substance Use Dilemmas – a Future Concern Despite a Positive Generational Effect?” aimed to identify the factors who may influence opinions concerning Australian medical students’ self-care and substance use behaviors as a means of providing insights into how future doctors view these issues compared to Australian doctors and the public. Overall, this is a relevant topic in the field, and it requires more works and efforts in this area to better understand how future doctors in Australia view alcohol misuse, illicit drug use, and cannabis use. Even though authors identified the opinion differences between the medical students, doctors, and the public, this difference could be confounded by other factors. The authors need conduct further analyses to assess the difference in opinions. Areas could be improved are described below:   

1.    Table 3: The authors compared the statistical significance difference in the responses between students, doctors, and public in Table 3. Because there are a total of three groups, it’s suggested to use Chi-square test to perform a global test to see whether there is a statistical significance among the whole three groups. If the purpose is to do a multiple comparison testing which is shown in the current table, a correction method should be used to compensate for Type I error. Using a P value of 0.05 to judge a significance is not appropriate since there are a total of three groups of participants. To sum up, it’s suggested to perform a global test first and then do the multiple comparisons using a correction method for the testing.

Further, the table 3 only shows univariate results. The differences in the opinions between medical students, doctors, and the public could be confounded by demographic factors. Even though the survey only collects data in age and gender, it’s suggested that authors conduct a multivariable model comparing the opinions adjusting for age and gender. With that saying, a multivariable logistic model should be conducted adjusting for age and gender. The odds ratio will show whether the observed statistical difference still exist after adjusting for age and gender.  

Author Response

Please see attachment and version 2 of the paper.

Reviewer 2 Report

The present study sought to analyze differences in the opinions related to substance use and self-care in three samples: medical students, graduated physicians, and general population. I believe that this study is interesting because it has several public health implications. Here are my overall comments:

Introduction

I see that three from the four questions that are intended to respond are centered around medical students. However, there is little emphasis about how self-care and substance use beliefs and opinions are relevant in particular for this group. I think that the health of future physicians is a relevant subject, however, the focus in the medical education curricula and the impact of current belief in the future practice should be further discussed. I think that the introduction rush into conclusion and fails to highlight several implications that regarding the subject that have been studied in the past 20-30 years.

Methods

Taking into consideration that the sources to reach each of the groups are quite different I believe that authors should describe with more detail the selection process. Please add a brief summary of the sampling procedures for POMS-I study. Also, it seems that a part of the medical doctor sample was recruited through social media, however, it is unclear how it was avoided that non-doctors respond the survey. Additionally, I believe that in the general public it becomes necessary to control for unique responses, however no method for such purpose is reported.

I have two concerns regarding the statistical analysis.

1)      Author used only bivariate statistics; therefore they assume that there are no confounders. Is important to notice that the sources of recruiting and other characteristics may render the samples incomparable (they are very different). Therefore, a method to control confounders is needed.

2)      I can more than 10 test of hypothesis that were conducted. However, correction for multiple comparisons was not mentioned.

Discussion

One important factor that is needed to be taken in consideration when comparing medical students vs graduated physicians is that there could be a cohort effect, since students will always be younger that graduates. This has several implications in their opinions and therefore must be discussed.

There are more limitations that the sampling method, and also is important to note that when a sample is highly biased, it can´t be corrected just by increasing its size. Another potential limitation that is worth to mention is that there are some characteristics that may turn the samples incomparable, for instance, the exposure to hospital settings is something that may change opinions and it doesn’t happen most of the times for general population.

An additional limitation it was that substance use or intention to use substances weren´t measured. This study only measured opinion about substance use. And most of the discussion treat the results as if they were equivalent or even close to substance use. Such assertion is very imprecise.  

Author Response

Please see the attachment and V2 of the paper.

Round 2

Reviewer 2 Report

I believe that most of my previous concerns have been responded and I think that the manuscript in its current form has improved in quality.